# Automatic pairing of real-world stimuli in younger and older adults: An event-related potential study

Petia Kojouharova *, István Czigler, Boglárka Nagy, Zsófia Anna Gaál

Institute of Cognitive Neuroscience and Psychology, HUN-REN Research Centre for Natural Sciences, Budapest, Hungary

* kojouharova.petia@ttk.hu

## Abstract

Contextual information can influence object perception, even when it is irrelevant to the task. When two unrelated stimuli frequently occur in close temporal succession, they may become associated. This process might be stronger in older adults due to age-related decline in inhibitory control, leading to deeper processing of irrelevant context. Automatic associations and their violations can be studied using visual mismatch negativity (vMMN), an event-related potential component reflecting the detection of violations of regularity in unattended stimuli. In our study, younger ($n = 18$, $M = 21.2 \pm 2.1$ yrs) and older ($n = 17$, $M = 69.8 \pm 2.8$ yrs) adults viewed pairs of images presented in succession: a scene (forest or street) followed by an emotional face (happy or angry). One scene–emotional face combination occurred frequently, the other rarely, forming a contextual oddball sequence. All individual stimuli appeared equally often. Participants performed an unrelated colour-change detection task, ensuring that the scene–emotional face pairs were unattended. We expected a vMMN to the emotional face in rare pairings, especially in older adults if they formed stronger associations. A control oddball condition with only emotional faces verified automatic emotion discrimination. In the control condition, vMMN emerged for both deviant emotions (in various ranges within the 94–276 ms post-stimulus time window). However, no vMMN was observed in the scene-emotional face pair condition. Instead, younger adults showed an early posterior positivity (90–161 ms) to rare pairings, while older adults exhibited a later negativity (356–384 ms). These results suggest that task-irrelevant, thematically unrelated visual events can become associated through temporal proximity. However, violations of these associations evoke neural responses distinct from vMMN and vary across age groups, with older adults relying on later, potentially semantic-related mechanisms rather than early visual processes to detect the regularity.

**Editor:** Árpád Csathó, University of Pecs Medical School, HUNGARY

**Data availability statement:** The data underlying the results presented in the study are available from the Open Science Framework database (https://doi.org/10.17605/OSF.IO/UNDJA).

**Funding:** This research was supported by the Hungarian Scientific Research Fund (NKFIH), grant number OTKA K 132880 and OTKA K 145940, awarded to ZsAG. NKFIH website: https://nkfih.gov.hu/about-the-office. The funders had no role in study design, data collection and analysis, decision to publish, or preparation of the manuscript.

**Competing interests:** The authors have declared that no competing interests exist.

## Introduction

Task-irrelevant information such as context may implicitly link to a target stimulus, facilitating or impeding our performance on a task. When searching for a familiar pub in a town we have visited before, the spatial arrangement of landmarks can help us, while a recently demolished building might mislead us. Similarly, anticipating sequential events – such as noticing the gradual darkening of clouds and unconsciously expecting rain – can support our behaviour. While the neural mechanisms of spatial context in guiding behaviour are well-documented, much less is known about whether consecutive unattended stimuli become automatically associated. This may be particularly relevant for older adults, for whom certain cognitive functions decline with age, but contextual information and automatic processes could serve as compensatory mechanisms. In this study, we investigated whether an association can form between two thematically unrelated, consecutively presented visual events that are not attended, and examined the electrophysiological correlates of violating such an association and tested whether these processes differ between younger and older adults.

In real-world settings, it is rare for objects to appear without any spatial or temporal context. A large body of research has demonstrated that regular spatial context has a considerable effect on object identification (e.g., [1,2]). Similarly, temporal regularity is helpful in the processing of objects and their features. Applying neurophysiological methods in studies investigating the effects of expected (regular) vs. surprising (irregular) successive attended events, surprising events elicited increased BOLD activity (e.g., [3]). In ERP studies where face stimuli were used, unpredictable alternating stimuli elicited larger responses than predictable ones (e.g., [4]), underscoring the presence of an expectation effect. Temporal regularity facilitates object processing even when such regularities are task-irrelevant (i.e., outside the focus of attention). Electrophysiological results from both auditory and visual modalities show that stimuli violating temporal regularities (i.e., deviant events) elicit distinct event-related responses – namely, the auditory mismatch negativity (MMN, for a review see [5]) and the visual mismatch response (vMMR, or visual mismatch negativity, vMMN, when the response is negative, for reviews see [6,7]).

MMN and vMMN are usually investigated using the passive oddball paradigm. In the visual version of this paradigm, standard events are constructed from stimuli with identical features or from stimuli belonging to the same category, and deviant events violate this regularity by differing in features or category. According to the predictive coding framework, (e.g., [7,8]), standard stimuli establish a memory-based prediction of expected stimuli, and the MMN/vMMN reflects a cascade of processing triggered by the detection of a surprising deviant event.

In the present study, our aim was to investigate a more complex form of temporal regularity: a particular scene (either a forest or a street) consistently preceded and was paired with faces expressing a specific emotion (happiness or anger). However, the pairings were unbalanced: one combination (e.g., forest scenes with happy faces and street scenes with angry faces) occurred frequently and thus served as the standards, while the reverse combination (e.g., forest scenes with angry faces and street

scenes with happy faces) occurred rarely and served as the deviants. Together, these frequent and rare pairings formed a contextual oddball sequence (scene-emotional face pair oddball). Importantly, both the scenes and the emotional faces were task-irrelevant, meaning that any association formed between them would have occurred automatically. A vMMN-like response to these infrequent, reversed pairings would indicate the automatic detection of an unexpected event. Previous research has shown that rare conjunctions of two features (i.e., deviant objects) can elicit a vMMN response [9]. The present study goes a step further: here the deviant conjunction is distributed across two sequential, thematically distinct visual events. So, we compared event-related potentials (ERPs) elicited by face stimuli in the frequent pairings (e.g., expected happy face after the forest scene) versus the rare pairings (e.g., unexpected happy face after the street scene).

To examine potential age-related differences in automatic temporal association learning, we compared older and younger participants. Both the formation of automatic associations between events and the detection of violation of such associations are important for adaptive behaviour. While many cognitive processes show age-related decline (e.g., [10–14]), various types of implicit learning are thought to remain relatively preserved in older adults (but see [15,16]). For example, results from studies using the contextual cueing paradigm suggest that this type of automatic learning is preserved with age [17–20]. When impairments have been reported [21], they were often attributed to a slower learning rate in older adults [22,23], or to the fact that older adults tend to acquire only a subset of repeated contexts [23]. Moreover, Yao and colleagues [23] found that, when tested at least four weeks after initial exposure, older adults retained contextual memory more robustly than younger adults, who required re-learning. At the same time, the two age groups may rely on different mechanisms to perform the task. While the spatial contextual cueing effect was behaviourally similar across age groups, underlying neural mechanisms differed in Kojouharova et al.'s [18] experiment: in younger adults, the effect was driven by enhanced early attention and categorization processes, whereas in older adults, it was associated with enhanced late, response-related activity.

As a control, we also included a traditional passive oddball condition using the same emotional faces from the contextual oddball task, to examine whether participants automatically discriminated between the two facial emotions (emotional-face-only oddball. Facial emotions are frequently and successfully applied in vMMN research (e.g., [24–26]). However, to the best of our knowledge, no study has yet investigated vMMN responses to emotional faces in older adults. As with other types of stimuli, vMMN results show that the component is intact, though often diminished in older adults compared to younger adults; for example, for checkerboard stimuli [27], and disappearing events [28], or see Kremláček et al. [29] for a review. These results were interpreted as indicating that younger adults process the deviant stimuli more extensively. However, when the integration of successively presented fragments of pseudo-letters was required to detect deviance, with a longer delay between the members of the pairs, older adults performed better than younger adults [30]. This result suggests that older adults' memory representation of the fragments persists for a longer time.

In the present study, we anticipated the following outcomes. First, if scenes and emotional faces are automatically associated, ERPs should differ for faces in the frequently paired scene-emotional face combinations compared to the same faces appearing in rare pairings. Second, if older adults process context and unattended faces more effectively due to impaired inhibitory control, the association (i.e., context effect) should be more likely to develop, and ERPs related to unexpected faces should emerge with greater amplitude. Finally, we expected vMMN to be elicited in both age groups for both emotional expressions in the emotional-face-only oddball paradigm. However, further comparisons between age groups or emotional expressions were beyond the scope of the present study.

## Methods

### Participants

Twenty-six younger and twenty-three older adults participated in the experiment. The younger adults were recruited through a school cooperative, while the older adults were recruited from our own database. All participants received payment for their participation. Three older participants did not complete the experiment. The data of five younger and three

older participants were excluded from analysis because of noisy EEG data. Thus, the final sample consisted of 18 participants in the younger group (13 women, 1 transgender man, $M = 21.17$ years, $SD = 2.06$, 2 left-handed) and 17 participants in the older group (10 women, $M = 69.82$ years, $SD = 2.81$, all right-handed). All had normal or corrected-to-normal vision and no reported neurological or psychiatric disorders. The size of the sample was determined by using G*Power [31] and was based on a previous study of vMMN to emotional faces. In Stefanics et al. [26], vMMN was elicited posteriorly to deviant emotional (happy and fearful) faces in three post-stimulus time windows. For each time window, $\eta^2$ was reported as the effect size for the main effect of stimulus (standard or deviant): 70–120 ms: $\eta^2 = 0.17$ (for fearful faces only), 170–220 ms: $\eta^2 = 0.32$, and 250–360 ms: $\eta^2 = 0.39$. Thus, taking into account the lowest effect size of $\eta^2 = 0.17$, a sample of 12 participants is required for power = 80%, alpha = 0.05.

To exclude dementia-related differences between the age groups, we measured intelligence with four subtests of the Hungarian version of Wechsler Adult Intelligence Scale (WAIS IV [32]) representing the four major components: Similarities – verbal comprehension; Digit Span – working memory; Matrix Reasoning – perceptual reasoning; Coding – processing speed. The scaled scores (where the age-group average is 10) achieved by the younger group were as follows: Similarities: $M = 13.5$ ($SD = 2.03$); Digit Span: $M = 10.06$ ($SD = 3.14$); Matrix Reasoning: $M = 11.06$ ($SD = 1.78$); Coding: $M = 11.22$ ($SD = 1.36$). The older group achieved the following scores on the four subtests: Similarities: $M = 13.88$ ($SD = 2.19$); Digit Span: $M = 12.35$ ($SD = 2.83$); Matrix Reasoning: $M = 11.65$ ($SD = 2.99$); Coding: $M = 14.59$ ($SD = 2.59$).

### Ethics statement

The study was approved by the United Ethical Review Committee for Research in Psychology (Hungary). Data collection was conducted from 8th December, 2022 until 29th June, 2023. Written informed consent was obtained from all individual participants included in the study.

### Stimuli and procedure

A summary of the stimuli and the procedure can be found on Fig 1.

The participants performed a task while task-irrelevant stimulus pairs were presented in sequences on a grey background (RGB 0.5, 0.5, 0.5, 45.8 cd/m$^2$). The stimuli were presented on a 24" monitor (ASUS VG245, resolution 1920 × 1080 px, refresh rate 60 Hz) located at a viewing distance of 140 cm. The task-relevant stimulus was a continuously present coloured rectangle frame (measuring 496 × 334 px and 5.6° × 3.7° of visual angle, line width 10 px) centred on the screen. The frame switched colours between red (RGB 1, 0, 0, 42.4 cd/m$^2$) and green (RGB 0, 1, 0, 154.2 cd/m$^2$) randomly every 5–15 seconds, and the participant had to indicate the colour switch by pressing Space on a computer keyboard. Responses were recorded as correct only if they were within 800 ms after the colour change. Key presses outside this time frame were not recorded. The purpose of the task was to distract attention from the task-irrelevant stimuli. After each sequence the participant was given feedback about their percentage of correct responses and their reaction time.

In the first half of the experimental session the task-irrelevant stimuli were image pairs. The first stimulus of the pair was a photo of a forest or a street. The second stimulus was an emotional face displaying either happiness or anger. We used one forest scene photo and one street scene photo. Both photos (see Fig 1A) were chosen from Unsplash (https://unsplash.com/), and their size was 486 × 324 px (5.5° × 3.6° of visual angle). The emotional face images were obtained from photos from the FACES database [33]. There were four emotional faces for both emotions: a younger woman, a younger man, an older woman, and an older man (see Fig 1A). The size of the images was 259 × 324 px (2.9° × 3.6° of visual angle). The images were centred on the screen and presented within the coloured frame. The stimuli pairs were presented in oddball sequences of 80 pairs. In each sequence each image was presented an equal number of times, i.e., on their own, the context forest and street images and the emotional happy and angry faces had a 0.5 probability. However, there were frequent (standard) pairs with a 0.8 probability and rare (deviant) pairs with a 0.2 probability. Half of the participants saw the forest-happy face and street-angry face as the standard pairs and the forest-angry face and

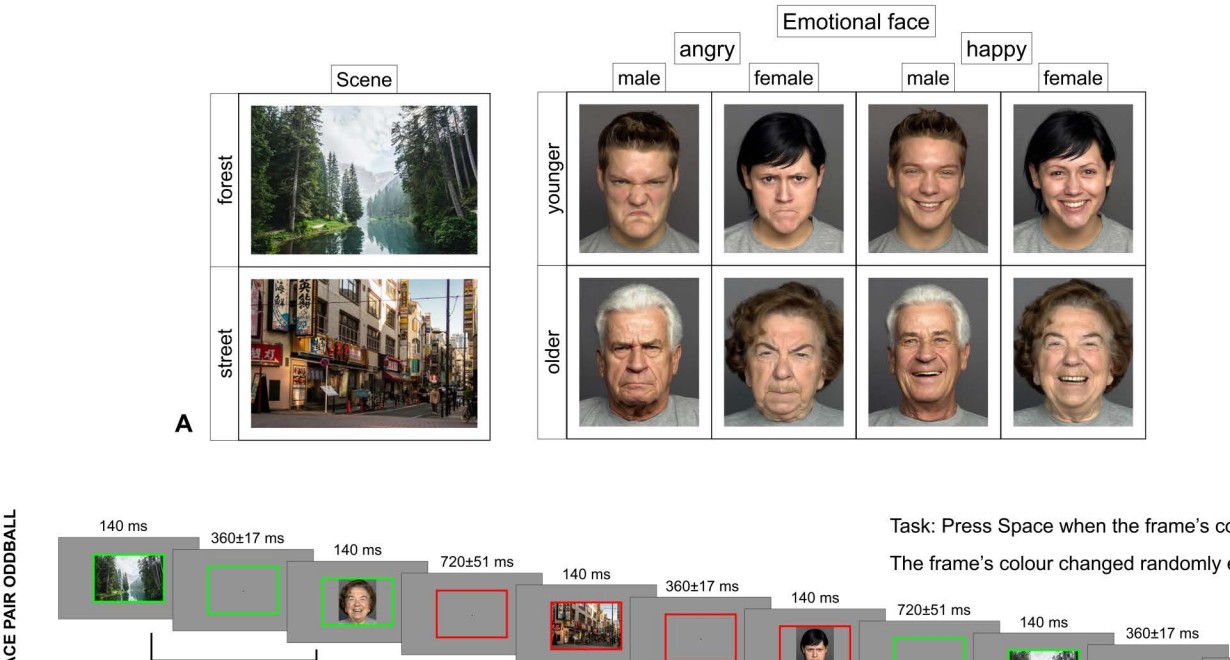

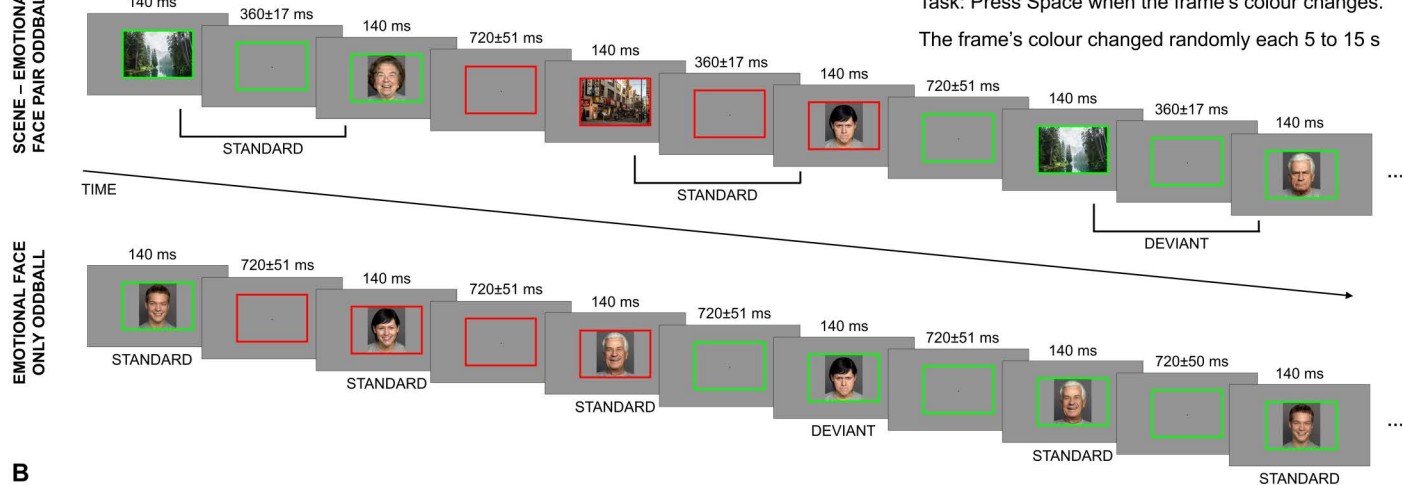

**Fig 1. Stimuli and procedure. (A)** Stimuli presented in the scene-emotional face pair oddball and in the emotional-face-only oddball. **(B)** Presentation of the stimuli in the scene-emotional face pair and in the emotional-face-only oddballs. The emotional face images presented on the figure were obtained from the FACES database [33].

street-happy face as the deviant pairs. For the other half of the participants the pairs' roles were reversed. A sequence always started with 6 standard pairs (3 of each). There were 10 sequences (presented in five blocks) with 64 standard pairs and 16 deviant pairs per sequence for a total of 640 standard pairs and 160 deviant pairs. Between two deviant pairs there were 2–6 standard pairs. Stimulus duration was 140 ms for both the scene and the emotional face, while the intra-pair interval was $360 \pm 17$ ms. The inter-pair interval was $720 \pm 51$ ms (in up to three steps of 17 ms). Each sequence was approximately 2 minutes long.

In the second part of the session only emotional faces were presented in oddball sequences. In half of the sequences the happy faces were the standard, and the angry faces were the deviant stimuli, and in the rest of the sequences the roles were reversed. In other words, we applied a reverse control where each emotional face was both a standard and a deviant. The order was counterbalanced between participants. The probability of the standard emotional face was 0.8, and the probability of the deviant emotional face was 0.2. A sequence always started with 6 standards, and there were

between 2 and 6 standards between deviants. There were 10 sequences (presented in five blocks) with 64 standard emotional faces and 16 deviant emotional faces in each sequence for a total of 320 standard happy faces, 80 deviant happy faces, 320 standard angry faces, and 80 deviant angry faces. The stimulus duration was 140 ms, and the interstimulus interval was 720±51 ms (in up to three steps of 17 ms). Each sequence was approximately 1.2 minutes long.

The stimuli presentation and response recording were realized using the Cogent 2000 [34] and Cogent Graphics [35] toolboxes in MATLAB [36].

### Recording and measuring of the electrophysiological activity

Brain electric activity was recorded (bandwidth: DC-70 Hz; sampling rate 1000 Hz; BrainVision actiCHamp amplifier, BrainVision Recorder, BrainVision Products GMBH) with Ag/AgCl active electrodes placed at 59 locations according to the extended 10–20 system using an elastic electrode cap (EasyCap, Brain Products GmbH). The reference electrode was placed on the nose. Eye movements were recorded with four electrodes placed around the eyes. Horizontal EOG was recorded with a bipolar configuration between electrodes positioned lateral to the outer canthi of the eyes (one electrode on each side). Vertical eye movement was monitored with a bipolar montage between two electrodes, one placed above and one below the left eye. The impedance of all electrodes was kept below 30 kΩ.

### Data analysis

**Behavioural analysis.** Mean accuracy (percentage of hits) and mean reaction time (RT, reaction time to the hits) were calculated for each sequence for a participant, and were then averaged for the two oddball conditions separately. A mixed analysis of variance (ANOVA) was conducted for both measures with ODDBALL (scene–emotional face pair or emotional-face-only) as the within-subject factor and AGE GROUP (younger or older) as the between-subject factor to investigate whether performance on the task differed depending on age and on the task-irrelevant stimuli. Slower RT was expected in the older compared to the younger group because of age-related slowing.

**ERP analysis.** The EEG signal was band-pass-filtered offline with a non-causal Kaiser-windowed FIR filter (lowpass filter parameters: cutoff frequency of 30 Hz, beta of 12.2653, a transition band of 10 Hz; highpass filter parameters: cutoff frequency of 0.1 Hz, beta of 5.6533, a transition band of 0.2 Hz). Stimulus-locked epochs were extracted for each emotional face in both conditions with a duration of 900 ms, including a 100 ms pre-stimulus interval.

In the case of the scene-emotional face pair oddball, standard epochs were the ones for the emotional faces that were expected after the scene image (i.e., from the standard pairs), and deviant epochs were the emotional faces that were unexpected after the scene image (i.e., from the deviant pairs). To equate the number of epochs for standards and deviants, only emotional faces from the standard pairs preceding deviant pairs were used for averaging. Because brain activity from the preceding (scene) stimulus did not return to baseline within the allotted 500 ms to allow for the use of the 100 ms pre-stimulus interval as a baseline, the 700–800 ms post-stimulus interval was used as a baseline instead.

In the case of the emotional-face-only oddball, epochs were extracted for the standard happy, standard angry, deviant happy, and deviant angry emotional faces. To equate the number of epochs for standards and deviants, only standard emotional faces preceding a deviant emotional face were used for averaging. Here, the 100 ms pre-stimulus interval served as the baseline.

The monitor lag (the time difference between the trigger and the appearance of a stimulus on the screen) was measured by a photodiode before the experiment. The lag was the same across events (67 ms), and thus the latencies of the events were corrected in the data after data collection. Epochs with an amplitude change exceeding 100 μV on any posterior (parietal, parietooccipital, or occipital) channel were excluded from further analysis. The average number of epochs for each analysis can be found in Table 1.

In the case of the scene-emotional face pair oddball, because our focus was on the second stimulus in the pair being differing from what should be expected, the epochs were pooled together independent of emotion (happiness and anger)

**Table 1. Average number of epochs with standard deviations in parentheses and ranges in the second row for deviant and standard stimuli preceding the deviants in each group and condition.**

| | | Scene-emotional face pair oddball | Emotional-face-only oddball | |
| --- | --- | --- | --- | --- |
| | | Emotional face | Happy | Angry |
| Younger group | Deviant | 132.4 (9.4)<br>109–145 | 66.4 (5.8)<br>56–76 | 65.8 (10.5)<br>44–98 |
| | Standard | 128.5 (11.5)<br>109–148 | 68.3 (9.3)<br>55–99 | 67.5 (4.8)<br>56–76 |
| Older group | Deviant | 131.3 (12.3)<br>95–143 | 62.7 (8.9)<br>34–71 | 65.1 (8.2)<br>42–73 |
| | Standard | 132 (14.9)<br>86–145 | 65 (8.7)<br>41–73 | 65.1 (9.6)<br>35–74 |

or preceding scene (forest or street), separately for standards and deviants, and were then averaged. A difference potential was then computed by subtracting the ERP for the standards from the ERP for the deviants.

In the emotional-face-only oddball, for each participant the epochs were averaged separately for the happy standard faces, deviant happy faces, angry standard faces and deviant angry faces. Then difference potentials were computed as a deviant-minus-standard ERP separately for happy (happy deviant minus happy standard) and angry (angry deviant minus angry standard) emotional faces. The purpose of this analysis was to show that vMMN is elicited for the emotional faces used in this study, in line with previous vMMN studies.

For all difference potentials we conducted point-by-point one-sample $t$-tests to 0 over the 50–400 ms post-stimulus time window (which was slightly longer that the expected 100–350 ms post-stimulus time window of vMMN) at the posterior regions of interest (ROI). This analysis is a necessary step before a direct comparison between the groups can be performed for the following reasons: 1) we need to establish the presence of vMMN within each group, and 2) if vMMN is observed, we need to establish whether it is observed in different ranges in each group – in older adults components may be in a different range (usually delayed) compared to younger adults. The following ROIs were formed: occipital (O1, Oz, O2), parietooccipital (PO3, POz, PO4), left parietooccipital (PO7, P7, P5), and right parietooccipital (PO8, P8, P6). The occipital and parietooccipital electrode sites are the expected sites of vMMN, while the left and right parietooccipital electrode sites are usually implicated in the processing of face stimuli, including in vMMN studies. At least twenty consecutive statistically significant negative differences (20 ms duration) at any of the four ROIs within the 50–400 ms post-stimulus time window were considered to show the elicitation of vMMN (cf. [37]).

For any such interval observed in either of the age groups in the scene-emotional face pair condition, we calculated mean amplitude, and then performed a repeated measures ANOVA with ROI (occipital, parietooccipital, left parietooccipital, right parietooccipital) as a within-subject factor and GROUP (younger, older) as a between-subject factor.

EEG data were preprocessed with MATLAB R2019b [38] and EEGLAB 2020.0 and 2024.0 [39]. JASP [40] was used for the analysis of the behavioural results.

## Results

### Behavioural results

Mean accuracy and RT are displayed in Table 2. There was a main effect of AGE GROUP on performance, $F(1,33) = 6.289$, $p = 0.017$, $\eta^2_p = 0.16$, with lower performance in the older group. The main effect of ODDBALL, $F(1,33) = 1.806$, $p = 0.188$, $\eta^2_p = 0.052$, and the interaction, $F(1,33) = 1.046$, $p = 0.314$, $\eta^2_p = 0.031$, were not significant. While the older adults had lower accuracy in the task than the younger adults, their performance was still above 90% which suggests adequate task engagement. The older adults were overall slower than the younger adults, $F(1,33) = 6.09$, $p = 0.019$, $\eta^2_p = 0.156$.

**Table 2. Mean accuracy and mean reaction time (RTs) for each group and condition.**

| | Mean accuracy (SD) in percentage | | Mean RT (SD) in ms | |
| --- | --- | --- | --- | --- |
| | Scene-emotional face pair oddball | Emotional-face-only oddball | Scene-emotional face pair oddball | Emotional-face-only oddball |
| **Younger group** | 97.75% (0.04) | 97.46% (0.02) | 440.2 (35.3) | 420.5 (48.3) |
| **Older group** | 94.3% (0.07) | 92.15% (0.08) | 479.7 (48.5) | 459.5 (58.4) |

Furthermore, both groups responded faster in the emotional-face-only oddball condition than in the scene-emotional face pair oddball condition, $F(1,33) = 14.394$, $p < 0.001$, $\eta^2_p = 0.304$. The interaction was not significant, $F(1,33) = 0.003$, $p = 0.96$, $\eta^2_p < 0.001$. Overall slower RTs are the norm for older adults compared to younger adults when performing the same task. An additional analysis with BLOCK (10 Blocks) as a within-subject factor and GROUP as a between-subject factor revealed a main effect of BLOCK, $F(3.945,130.186) = 6.05$, $p < 0.001$, $\eta^2_p = 0.155$, but no BLOCK×GROUP interaction, $F(3.945,130.186) = 1.457$, $p = 0.22$, $\eta^2_p = 0.042$, suggesting that the difference between the oddball conditions was due to an improvement in performance during the session as the oddball conditions were always presented in the same order.

## ERP results

Figs 2–4 show the ERPs and the difference potentials in both in the scene-emotional face pair oddball and in the emotional-face-only oddball conditions. Latency ranges with significant difference (20 consecutive difference at significant *t*-tests at a posterior ROI) are marked in the difference potentials. Scalp distributions in these ranges are also shown on Figs 2–4. As the deviant-minus-standard difference potentials indicates, in the scene-emotional face pair oddball condition in the younger group the only difference emerged in the 90–161 ms range, as a positivity. In the older group a negative difference appeared in the 356–384 ms range. Table 3 shows the ranges in the various ROIs.

The difference potentials differed significantly between the groups in the 90–161 ms time interval, $F(1,33) = 4.583$, $p = 0.04$, $\eta^2_p = 0.122$. There was no main effect of ROI, $F(3,99) = 1.598$, $p = 0.195$, $\eta^2_p = 0.046$, and no interaction, $F(3,99) = 2.078$, $p = 0.108$, $\eta^2_p = 0.059$. There were neither ROI, $F(2.217,73.172) = 0.556$, $p = 0.645$, $\eta^2_p = 0.017$, nor GROUP main effects, $F(1,33) = 0.666$, $p = 0.42$, $\eta^2_p = 0.02$, for the 356–384 ms time interval. The ROI×GROUP interaction was significant, $F(2.217,73.172) = 6.646$, $p = 0.002$, $\eta^2_p = 0.168$. The *post hoc* tests revealed only a significant difference in mean amplitude between the parietooccipital and the right parietooccipital ROI in the younger group ($p = 0.016$), with the amplitude being more negative in the right parietooccipital ROI.

In the emotional-face-only condition negative deviant-minus-standard difference appeared in various ranges within 144–276 ms in the younger group, and within 94–269 ms in the older group. Details of the ranges in the various ROIs and for happy and angry deviants are shown on Table 3. The negative difference can be considered as vMMN.

## Discussion

In this study, we investigated whether participants automatically form associations between two temporally linked, thematically unrelated visual events – specifically, a scene and an emotional facial expression – when these events are not task-relevant. We also examined whether violations of these associations elicit distinct electrophysiological responses, and whether these responses differ between younger and older adults.

In the traditional oddball design using happy and angry faces, we observed ERP differences between deviant and standard stimuli: at posterior sites, deviant-related ERPs were more negative than those elicited by standard stimuli. These differences were comparable across the two age groups. In the younger group, deviant happy faces elicited only short negativity (9–16 ms), however, it was elicited at all ROIs. This result is putatively indicating the sensitivity to violated regularity of repetitive emotional category, i.e., the emergence of vMMN (e.g., [24–26]).

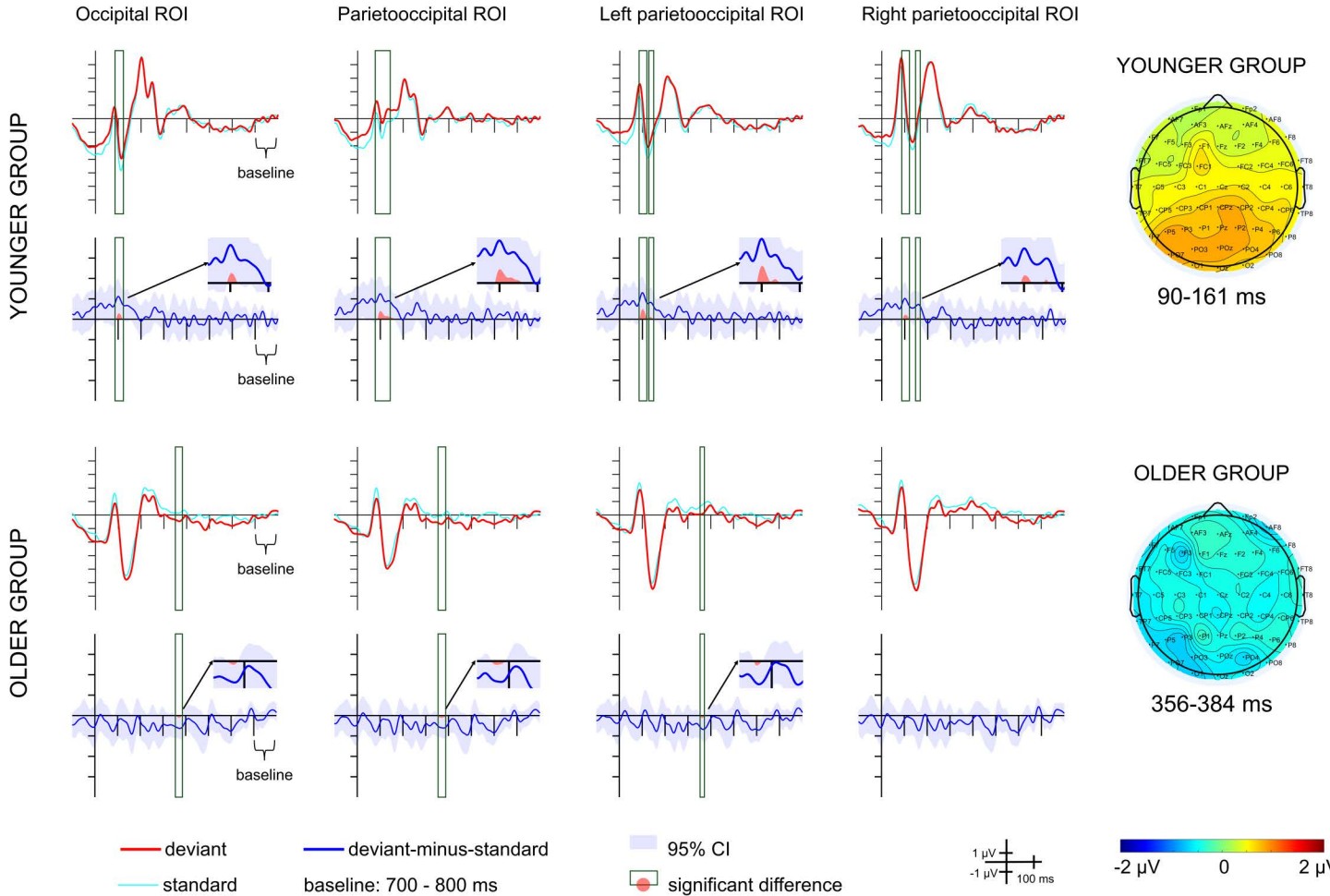

**Fig 2. Average ERPs and deviant-minus-standard difference potentials for each group and region of interest (ROI) in the scene-emotional face pair oddball condition.** Red lines depict ERPs to the deviant stimuli, light blue lines depict ERPs to the standard stimuli preceding the deviants, blue lines depict the deviant-minus-standard difference potentials, with the 95% CI interval shown by the blue shaded area. Scalp distributions for the ranges in which significant differences were found are shown on the right.

Contrary to the results of the traditional oddball paradigm, no unambiguously vMMN-related activity emerged in the scene-emotional face pair oddball condition. Thus, we found no evidence for the engagement of similar mechanisms when the regular pairing of scene and facial emotion was violated by a rare, deviant pairing – that is, when the face was preceded by the alternative scene. However, we cannot conclude that the regular contingency remained undetected. In the younger group, the deviant facial emotion elicited an early positivity, while in the older group, a later negativity. The emergence of early posterior positivity in response to stimuli that differ from the first element of a pair is a well-documented phenomenon, even when the varying visual features are task-irrelevant (e.g., [41]–[43]). A novel finding of the present study is that a similar early positivity can also be elicited by physically different, but regularly associated, visually complex events. As an age-related difference, we recorded no such early effect in the older group. However, in a later ERP time window, older adults also showed sensitivity to the violation of the association between the scene and the facial expression. One study [44] found a difference to deviant faces of old models among standard faces of young models in the 330–370 ms range in older adults, which the authors interpreted

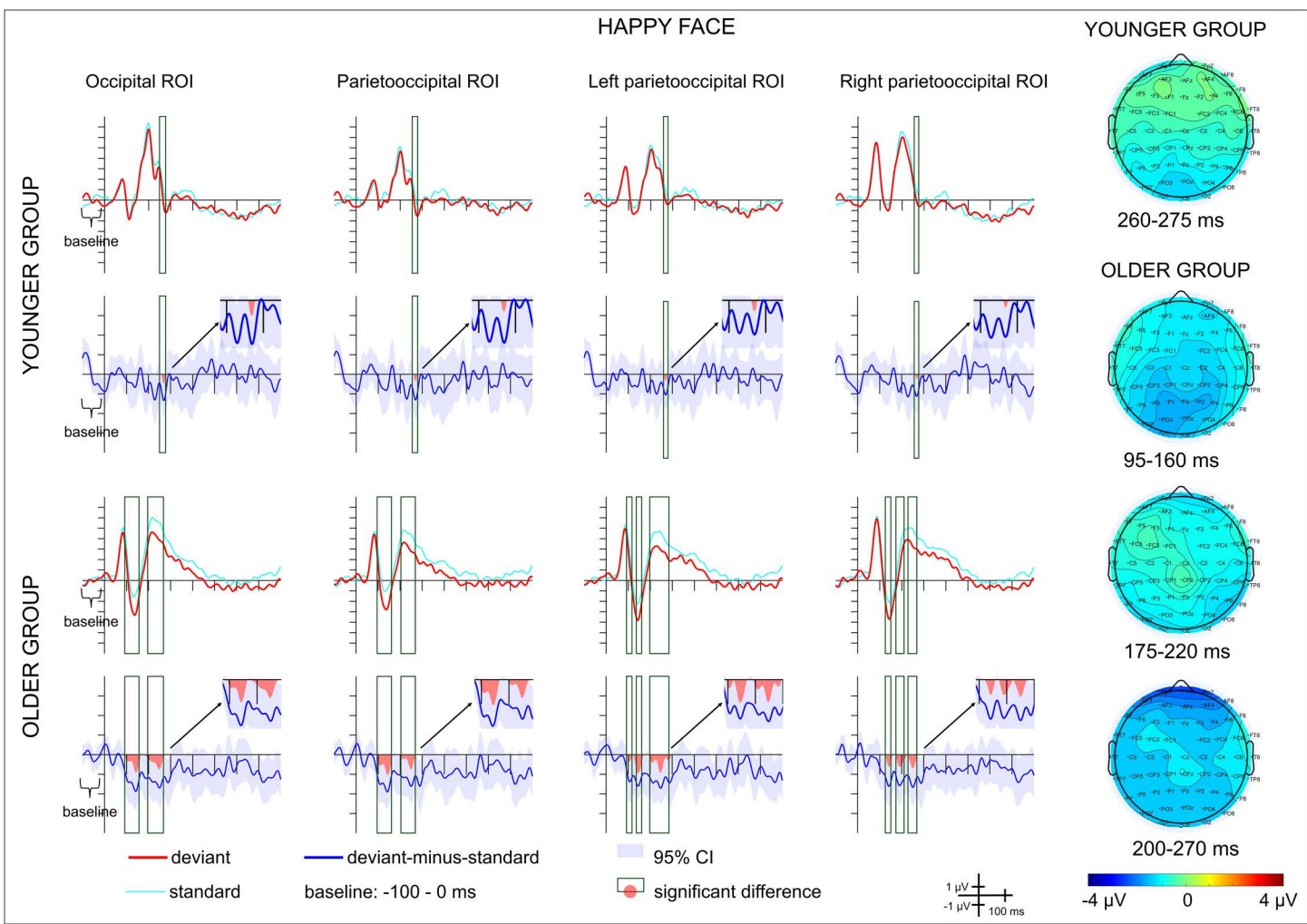

**Fig 3. Average ERPs and deviant-minus-standard difference potentials for each group and region of interest (ROI) in the happy emotional-face-only oddball condition.** Red lines depict ERPs to the deviant stimuli, light blue lines depict ERPs to the standard stimuli preceding the deviants, blue lines depict the deviant-minus-standard difference potentials, with the 95% CI interval shown by the blue shaded area. Scalp distributions for the ranges in which significant differences were found are shown on the right.

as a delayed vMMN. However, this interpretation remains debated, as responses in this latency range may reflect higher-level evaluative processes rather than classical mismatch detection. In the present study we did not observe a comparable delayed response to emotional faces in the emotional-face-only condition. One possibility is that violations involving more complex or abstract associations (e.g., age in [44], or cross-category associations in the present study) may not elicit a canonical vMMN in older adults, but instead engage later processing stages. At this stage of stimulus processing, it is likely that mechanisms other than predictive coding are involved. Indeed, the detection of incongruence in this latency range is well documented in other domains, most prominently in the N400 component, which is typically observed in response to unexpected verbal stimuli (for a review see [45]). Age-related differences in change detection under cognitively more complex conditions have also been observed using the reference-back working memory paradigm [46]. The study reported a late anterior positivity in younger adults, and an even later posterior positivity in older adults on certain types of deviant trials.

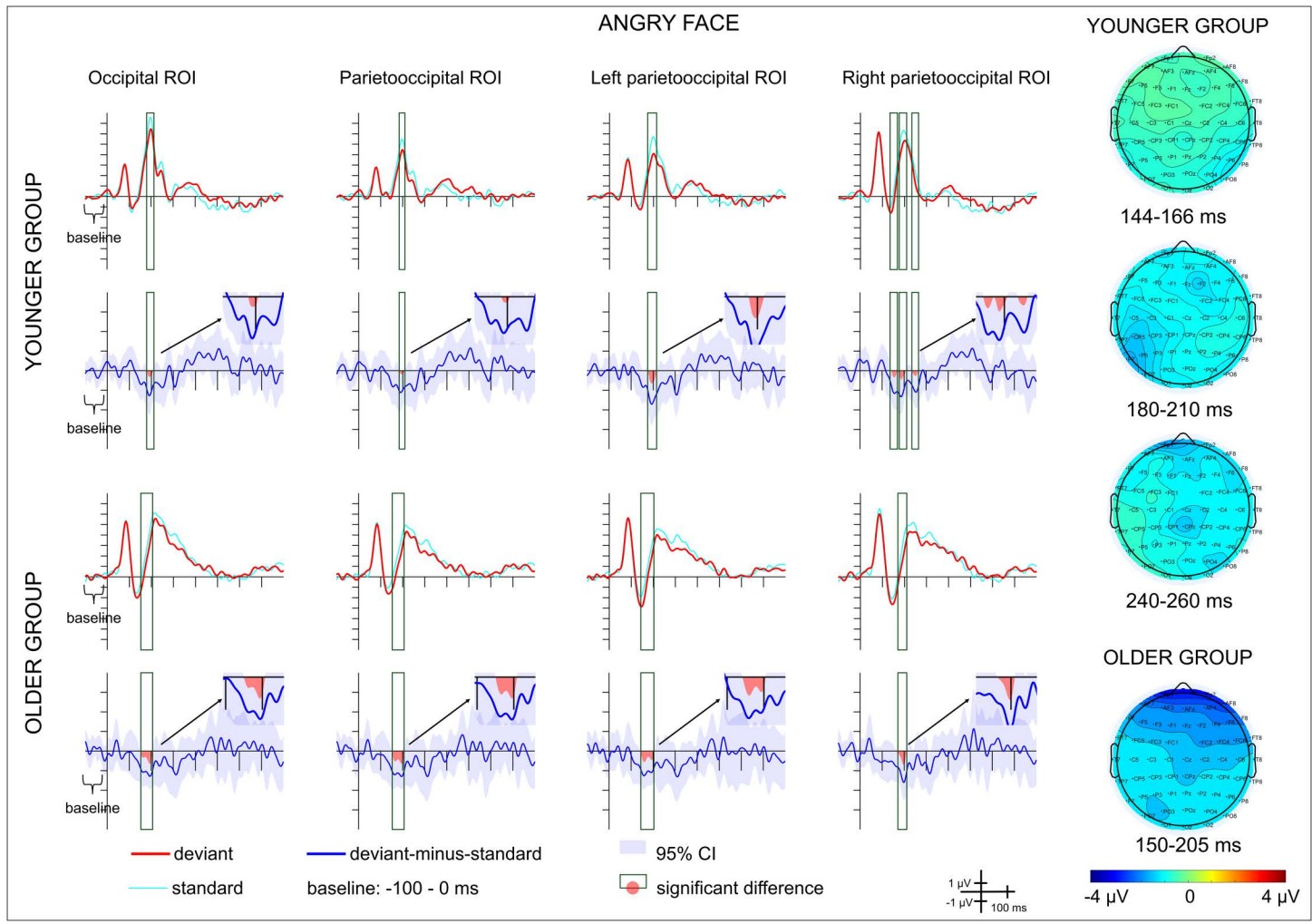

**Fig 4. Average ERPs and deviant-minus-standard difference potentials for each group and region of interest (ROI) in the angry emotional-face-only oddball condition.** Red lines depict ERPs to the deviant stimuli, light blue lines depict ERPs to the standard stimuli preceding the deviants, blue lines depict the deviant-minus-standard difference potentials, with the 95% CI interval shown by the blue shaded area. Scalp distributions for the ranges in which significant differences were found are shown on the right.

What are the fundamental differences between the traditional passive oddball paradigm – where vMMN typically emerges even when the deviant is a rare conjunction of features such as colour and grating orientation (e.g., [9]) – and the current paradigm, in which no vMMN was observed? The simplest explanation could be the lack of an association between the scene and the emotional face in the absence of task relevance or attention. However, this explanation is challenged by the early positivity observed in the younger group, and the later negativity observed in the older group, both indicating that the deviant pairing was processed differently. From the perspective of predictive coding accounts, one might expect vMMN-like activity to emerge, given that the deviant facial emotion was unexpected. However, according to more nuanced interpretations of the predictive theory (e.g., [7]), vMMN reflects a process involved in the identification – or attempted identification – of surprising events (whether successful or not, see [46]). In the scene-emotional face pairing condition, while the deviant emotional face was indeed unpredicted, it belonged to a stimulus category (emotional faces) presented with equal overall probability. Thus, its identification did not require the full cascade of processes hypothesised

**Table 3. Ranges of significant differences in the deviant-minus-standard difference potentials for each group, condition, and region of interest (ROI).**

| | | Occipital ROI | Parietooccipital ROI | Left parietooccipital ROI | Right parietooccipital ROI |
|---|---|---|---|---|---|
| **Younger group** | **Scene-emotional face pair oddball** | 94–116 ms | 91–151 ms | 90–119 ms, 129–141 ms | 94–115 ms, 155–161 ms |
| | **Happy emotional-face-only oddball** | 260–276 ms | 262–274 ms | 262–275 ms | 264–273 ms |
| | **Angry emotional-face-only oddball** | 182–205 ms | 187–204 ms | 179–215 ms | 144–166 ms, 177–206 ms, 239–260 ms |
| **Older group** | **Scene-emotional face pair oddball** | 362–378 ms | 358–384 ms, | 356–369 ms, | |
| | **Happy emotional-face-only oddball** | 95–162 ms, 194–269 ms | 94–164 ms, 201–268 ms | 98–117 ms, 125–167 ms, 197–291 ms | 127–156 ms, 174–220 ms, 227–269 ms |
| | **Angry emotional-face-only oddball** | 152–206 ms | 153–207 ms | 144–203 ms | 170–209 ms |

by predictive coding theory as necessary for eliciting vMMN. The early positivity has been interpreted as reflecting feature-specific, memory-based change detection, possibly related to the vMMN [41–43], i.e., in the younger group an early mismatch between expected features was sufficient to signal a violation of the association. Another important consideration is the nature of the manipulated regularity. VMMN in the context of complex stimuli remains largely unexplored (e.g., [47]). Unlike classical vMMN paradigms, the scene-emotional face pair condition involved category-level alternations between complex stimuli, which may not give rise to stable low-level predictions required for eliciting a classical vMMN. These alternations may engage higher-level categorical or semantic processing, not captured by early vMMN components. This may be reflected in the later negativity in older adults, where the association appears to have been encoded as a more complex regularity. This may explain the absence of a vMMN-like response in our data.

To summarize the present results, our findings suggest that frequent pairings of two thematically unrelated visual events can become associated, even when these events are not task-relevant. However, the ERP response to the violation of such associations differs from the neural mechanisms underlying the detection of these violations appear to differ between older and younger adults.

## Acknowledgments

We thank Krisztina Czinkóczi for her technical assistance.

## Author contributions

**Conceptualization:** Petia Kojouharova, István Czigler, Boglárka Nagy, Zsófia Anna Gaál.

**Data curation:** Petia Kojouharova.

**Formal analysis:** Petia Kojouharova.

**Funding acquisition:** Zsófia Anna Gaál.

**Investigation:** Petia Kojouharova.

**Methodology:** Petia Kojouharova, István Czigler, Boglárka Nagy, Zsófia Anna Gaál.

**Project administration:** Petia Kojouharova, Zsófia Anna Gaál.

**Software:** Petia Kojouharova.

**Supervision:** Zsófia Anna Gaál.

**Visualization:** Petia Kojouharova.

**Writing – original draft:** Petia Kojouharova, István Czigler, Zsófia Anna Gaál.

**Writing – review & editing:** Petia Kojouharova, István Czigler, Boglárka Nagy, Zsófia Anna Gaál.

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
