## [Decision Letter · Decision Letter 0]

30 Nov 2025

Dear Dr. Kojouharova,

Thank you for submitting your manuscript to PLOS ONE. After careful consideration, we feel that it has merit but does not fully meet PLOS ONE’s publication criteria as it currently stands. Therefore, we invite you to submit a revised version of the manuscript that addresses the points raised during the review process.

I apologize for the delay in the review process. The first two reviewers reached differing recommendations, which made it necessary to invite a third reviewer to provide an additional evaluation.

Reviewer 1 raised several critical comments regarding both the methodology and the analyses (and recommended rejection). In contrast, Reviewer 2 (recommended major revision) and Reviewer 3 (recommended minor revision) provided overall positive evaluations; their concerns focus primarily on the results section and the interpretation rather than on the methodological approach.

I encourage the authors to submit a revision. In the revised manuscript, please make sure to address the concerns raised by all three reviewers.

We look forward to receiving your revised manuscript.

Kind regards,

Árpád Csathó, Ph.D.

Academic Editor

PLOS ONE

Journal Requirements:

“This research was supported by the Hungarian Scientific Research Fund (NKFIH), grant number OTKA K 132880, awarded to ZsAG. NKFIH website: https://nkfih.gov.hu/about-the-office . NKFIH did not play any role at any stage of the study.”

4. We note that Figure 1 includes an image of a [participant / in the study].

Please respond by return e-mail with an amended manuscript. We can upload this to your submission on your behalf.

If you are unable to obtain consent from the subject of the photograph, please either instruct us to remove the figure or supply a replacement figure by return e-mail for which you hold the relevant copyright permissions and subject consents. In some cases, you may need to specify in the text that the image used in the figure is not the original image used in the study, but a similar image used for illustrative purposes only. We can make any changes on your behalf.

Reviewers' comments:

Reviewer's Responses to Questions

**Comments to the Author**

1. Is the manuscript technically sound, and do the data support the conclusions?

Reviewer #1: No

Reviewer #2: Partly

Reviewer #3: Yes

2. Has the statistical analysis been performed appropriately and rigorously?

Reviewer #1: No

Reviewer #2: I Don't Know

Reviewer #3: Yes

3. Have the authors made all data underlying the findings in their manuscript fully available?

Reviewer #1: Yes

Reviewer #2: Yes

Reviewer #3: Yes

4. Is the manuscript presented in an intelligible fashion and written in standard English?

Reviewer #1: Yes

Reviewer #2: Yes

Reviewer #3: Yes

Reviewer #1: This paper presents an experiment on young and older adults’ ability to ignore or inhibit irrelevant stimuli while attending to and performing another and different task. The authors designed an experiment in which they presented pairs of images successively; either a scene (forest or street) followed by an emotional face (happy or angry). One scene-emotional face (either happy or angry) occurred frequently, while the other was presented rarely. This procedure thus generated a contextual oddball sequence. The participant’s task was to detect a change in a colored square, thus creating a situation in which the scene-emotional face pairings were presumably unattended or ignored. The authors predicted that, because older adults are thought to have a deficit in inhibitory control, they would not be able to ignore the rare, scene-emotional face pairings and would, therefore, demonstrate a larger visual mismatch negativity than the younger adults. But, as noted below, according to the authors’ explanations, there was no difference between the young and older adults vMMNs.

The abstract does not state that there were no age group differences in the visual MMN although, as far as I can tell, these differences were not assessed by ANOVA, as was done for the behavioral data. Furthermore, the abstract does not conclude with an interpretation of what the age differences that did occur (a positivity for young adults and a negativity, interpreted as an N400, for the older adults) imply for aging-related information processing.

The design of the experiment seems straightforward, but I have a number of difficulties for the way in which the ERP differences were statistically analyzed and with the interpretation the authors offered for the differences (alluded to above) between the age groups.

The authors used an IQ test to determine whether any of their older participants were demented. This seems odd, as there are several reliable assessments of dementia-related illness that the authors could have used instead. Also, how did the authors rule out young and/or older adults who might have been suffering from mental illnesses such as depression and or bipolar disorder among a few other mental disorders?

Rather than analyzing the ERP data as had been performed on the behavioral data using ANOVAs in which the effects of age group could have been assessed, the authors analyzed the ERP data using t-tests from zero to assess whether a vMMN was present. Because the authors made a strong prediction that older adults would, due to difficulties in inhibitory control, show larger vMMN than young adults, these data too should have been assessed via ANOVAs using averaged voltages over the time period when a vMMN would have been expected. Instead, the statistical test for the presence of a vMMN in both age groups was the presence of 20 sequentially significant t-tests from zero (at an alpha level of 0.05) in the time range when the vMMN was present. Using this technique creates a problem, in which statistical differences could occur by chance. Hence, some correction for this possibility must be included. Although the Bonferroni correction is a conservative test, it has been used in these situations. Thus, 20 tests at an alpha level of 0.05, would require a significant difference at 0.0025 (0.05/20). Unfortunately, the authors do not mention what the alpha levels at each of the 20 points were. In fact, inspection of the ERP data in the figures, shows that the negativities the authors point to are quite small, again suggesting a strict criterion for significance. The authors might want to point with arrows specifically to the negativities they did analyze. Additionally, as can be seen in the scalp maps, there is very little focal activity over the posterior scalp where the vMMN might have been expected. Similarly, the other areas of the scalp also show very small variations in focal activity.

Finally, the Discussion is meager. There is very little discussion of (what I see) as a lack of difference in the vMMNs between age groups. Moreover, the discussion of the age-group differences that did occur is extremely limited and, of necessity, completely post-hoc. For example, I do not see why an N400 would be expected in the oddball situation presented in the paper. Yet the authors do very little by way of telling us why an N400 might have occurred and what type of cognition that might entail for the older relative to the younger adults who produced an early positivity. The authors do little in the way of interpreting these age-related differences. Why would they occur and what do they mean in terms of the differences in cognition (if they exist here) between the young and older adults?

On another note, it might have been useful to determine if there was any association, across participants, between the behavioral data and the size of the vMMN, the positivity in young adults and the "N400" in older adults. These associations, were they present and differed between young and older adults, might have aided the authors in coming up with a more definitive explanation for the age-related differences that did occur.

Reviewer #2: Thank you for this paper. It is very nicely written with all the details required to repeat the investigation. Thursday introduction and methods section do not need any changes, except possibly a reference on the method to discover significant changes between the two groups (i.e. consecutive t-tests).

The results section has two figures showing the ERPs for standard and deviant stimuli.

Fig2 shows the contextual MMNs. The older group does not have a plausible MMN looking at the figure. Baseline is ok and hence one should see an MMN if it was there. The younger group do have an MMN but the baselines of the ERPs are not aligned which I think causes the difference to show up on the MMN potentials.

Fig 3 shows almost identical ERPs for happy and angry faces for the young. The MMN looks more like noise for each of the ROIs. The older populations do have a change that is easily appreciated by the reader and hence also a MMN. Note also that the baselines are close to 0 and bit horizontal at the start of the ERPS which I think will allow you to see a possible MMN. The older group do have an MMN but the younger do not, which is strange.

I presented figures do not show a MMN for either gourd for the contextual data and not for the faces only for the young group.

Is there any other ROIs that could be used to enhance the data to create MMNs that look reproducible? At the moment I feel that these 2 figures are key to the paper. I am worried that the numbers used (receptions of images and number of patients) might be two small.

Reviewer #3: In this study, the authors investigated neural responses to violations of visual temporal regularity in young and older healthy adults. A contextual oddball task with scene-face pairings was used to test this effect. A traditional oddball task was also included to examine established visual mismatch negativity (vMMN) responses to deviant stimuli. vMMN activity was detected in both age groups in the traditional oddball task. Such activity did not emerge in the scene-face sequence task, although the two groups showed distinct neural responses when processing deviant stimuli in this task.

Overall, this is a solid and relevant study worthy of publication. Nonetheless, I have a few questions I hope the authors could clarify:

(1) I found the title a bit misleading. What makes the stimuli “complex”? Perhaps “sequential visual stimuli” would be a better fit for this study.

(2) Did the authors test differences in ERP results between the two age groups? This seems to be an important point of their study, but I could not find an analysis of this effect.

(3) How did behavioral and ERP results relate to one another? An explanation for the group differences in response accuracy and reaction time, as well as an interpretation of how these differences relate to the neural responses, should be provided in the Discussion.

(4) Could you elaborate further on why vMMN responses were not detected in the scene-face oddball task? While predictive coding is clearly relevant for framing the results of the traditional oddball task, I found the interpretation of the neural responses to the scene-face task a bit scarce. For example, these results could be linked to studies using other sequential stimuli, such as music (e.g., Bonetti et al., 2024a, 2024b).

Bibliography:

Bonetti, L., Fernández-Rubio, G., Carlomagno, F., Dietz, M., Pantazis, D., Vuust, P., & Kringelbach, M. L. (2024a). Spatiotemporal brain hierarchies of auditory memory recognition and predictive coding. Nature Communications, 15(1), 4313. https://doi.org/10.1038/s41467-024-48302-4

Bonetti, L., Fernández-Rubio, G., Lumaca, M., Carlomagno, F., Risgaard Olsen, E., Criscuolo, A., Kotz, S. A., Vuust, P., Brattico, E., & Kringelbach, M. L. (2024b). Age-related neural changes underlying long-term recognition of musical sequences. Communications Biology, 7(1), 1036. https://doi.org/10.1038/s42003-024-06587-7

**Do you want your identity to be public for this peer review?** For information about this choice, including consent withdrawal, please see our Privacy Policy

Reviewer #1: No

Reviewer #2: **Yes:** Gerald K. Cooray

Reviewer #3: No

---

## [Author Response · Author response to Decision Letter 1]

21 Jan 2026

We thank the reviewers for the careful consideration of our manuscript and for their constructive and detailed comments. All comments have been addressed in detail below, and corresponding changes have been made in the revised manuscript. Reviewer comments are marked with grey.

Reviewer #1:

This paper presents an experiment on young and older adults’ ability to ignore or inhibit irrelevant stimuli while attending to and performing another and different task. The authors designed an experiment in which they presented pairs of images successively; either a scene (forest or street) followed by an emotional face (happy or angry). One scene-emotional face (either happy or angry) occurred frequently, while the other was presented rarely. This procedure thus generated a contextual oddball sequence. The participant’s task was to detect a change in a colored square, thus creating a situation in which the scene-emotional face pairings were presumably unattended or ignored. The authors predicted that, because older adults are thought to have a deficit in inhibitory control, they would not be able to ignore the rare, scene-emotional face pairings and would, therefore, demonstrate a larger visual mismatch negativity than the younger adults. But, as noted below, according to the authors’ explanations, there was no difference between the young and older adults vMMNs.

- We thank the reviewer for the detailed and constructive comments, which helped us clarify ambiguous points and address gaps in the manuscript!

1. The abstract does not state that there were no age group differences in the visual MMN although, as far as I can tell, these differences were not assessed by ANOVA, as was done for the behavioral data. Furthermore, the abstract does not conclude with an interpretation of what the age differences that did occur (a positivity for young adults and a negativity, interpreted as an N400, for the older adults) imply for aging-related information processing.

- We added the following text to the abstract:

“However, violations of these associations evoke neural responses distinct from vMMN and vary across age groups, with older adults relying on later, potentially semantic-related mechanisms rather than early visual processes to detect the regularity.”

We respond in connection with the ANOVA comparison below.

2. The design of the experiment seems straightforward, but I have a number of difficulties for the way in which the ERP differences were statistically analyzed and with the interpretation the authors offered for the differences (alluded to above) between the age groups.

- We hope that the response to the relevant questions below clarified the issues.

3. The authors used an IQ test to determine whether any of their older participants were demented. This seems odd, as there are several reliable assessments of dementia-related illness that the authors could have used instead. Also, how did the authors rule out young and/or older adults who might have been suffering from mental illnesses such as depression and or bipolar disorder among a few other mental disorders?

- Assessments such as MoCA and MMSE provide a brief global screening of cognitive status and are commonly used in clinical contexts. In our lab, however, we routinely administer the WAIS-IV, as it allows us to assess potential decline in specific cognitive domains rather than providing only a general cognitive score. This is particularly relevant given that several of our studies aim to relate individual differences in specific cognitive functions to electrophysiological processes. For reasons of methodological consistency, we therefore apply the same cognitive assessment protocol for all studies, even in cases where the detailed cognitive measures are not included in the final analysis. Importantly, the WAIS-IV is not used as a diagnostic tool for dementia, but as a comprehensive assessment of cognitive functioning.

With respect to psychiatric conditions, participants could only be excluded based on self-report. This approach is a standard practice in cognitive and electrophysiological research, as comprehensive clinical assessment of psychiatric disorders is typically not feasible due to resource constraints. Our recruitment notices explicitly state exclusion criteria regarding neurological diseases, and participants were required to confirm this during the informed consent process. The informed consent form included the following statement (here translated from Hungarian):

“I declare that I do not suffer from any neurological disease, and that I do not take stimulants or sedatives, nor any other medication affecting the central nervous system on a long-term basis.”

4. Rather than analyzing the ERP data as had been performed on the behavioral data using ANOVAs in which the effects of age group could have been assessed, the authors analyzed the ERP data using t-tests from zero to assess whether a vMMN was present. Because the authors made a strong prediction that older adults would, due to difficulties in inhibitory control, show larger vMMN than young adults, these data too should have been assessed via ANOVAs using averaged voltages over the time period when a vMMN would have been expected. Instead, the statistical test for the presence of a vMMN in both age groups was the presence of 20 sequentially significant t-tests from zero (at an alpha level of 0.05) in the time range when the vMMN was present. Using this technique creates a problem, in which statistical differences could occur by chance. Hence, some correction for this possibility must be included. Although the Bonferroni correction is a conservative test, it has been used in these situations. Thus, 20 tests at an alpha level of 0.05, would require a significant difference at 0.0025 (0.05/20). Unfortunately, the authors do not mention what the alpha levels at each of the 20 points were. In fact, inspection of the ERP data in the figures, shows that the negativities the authors point to are quite small, again suggesting a strict criterion for significance. The authors might want to point with arrows specifically to the negativities they did analyze. Additionally, as can be seen in the scalp maps, there is very little focal activity over the posterior scalp where the vMMN might have been expected. Similarly, the other areas of the scalp also show very small variations in focal activity.

- We appreciate the detailed comment, which highlighted areas that were insufficiently elaborated in the manuscript and, hopefully, helped us provide greater clarity! We respond to all analysis issued below.

Regarding a comparison between the groups, there are two points that we think should be considered here. One is the morphology of the difference potential within a group. When a possible component is visible in the difference potential (i.e. the difference potential differs from 0 for shorter or longer intervals), the consecutive t-tests to 0 show us whether the noise in the data within the groups is larger than the signal. If not (i.e. if the t-tests are significant), we assume the presence of a component. The morphology, however, can be very different between the groups as seen in the present study – there is no indication of a positive component in the early range in the older group and virtually no indication of a negative component in the late range in the younger group. In other words, a direct comparison would compare the noise of a component in one group to the noise of no component in the other group. If the noise is larger in the group in which the component is not observed in the grand average and the groups are not significantly different, this does not invalidate the presence of the difference within the group in which it is observed. So, it is important to establish whether vMMN is observed at all within the group. The second point is that even if a component is observed in both groups, it may be in different ranges – ERP components are often delayed in older adults. The test for the presence of vMMN is also informative as to which ranges should be compared for a difference in amplitude, and whether the ranges may be too different for a meaningful comparison.

We have now added the following text to the Data Analysis section:

“This analysis is a necessary step before a direct comparison between the groups can be performed for the following reasons: 1) we need to establish the presence of vMMN within each group, and 2) if vMMN is observed, we need to establish whether it is observed in different ranges in each group – in older adults components may be in a different range (usually delayed) compared to younger adults.”

However, we agree that a direct comparison could be informative – a significant difference between the groups could indicate a larger effect, i.e. large enough to be observed above the noise in the data. We have now included an ANOVA analysis for the two ranges in the scene – emotional face pair oddball in the relevant sections under Methods and Results. The ANOVA shows a significant difference between the groups in the early but not in the later range.

Regarding the consecutive t-tests as a criterion, an ERP (or a difference potential) consists of many consecutive time points which are not independent of each other, i.e. a difference potential consists of autocorrelated data and the presence of multiple consecutive t-tests are a strong indicator of the presence of a component. The Bonferroni correction assumes that the data are independent, and while it can be used for autocorrelated data, it is too conservative. Our approach is based on Guthrie & Buchwald (1991, https://doi.org/10.1111/j.1469-8986.1991.tb00417.x) which suggests that the criterion should be based on the number of significant consecutive t-tests. We have now included the reference in the manuscript. Note that there is no single, accepted criterion for establishing the presence of vMMN in existing vMMN literature, different labs rely on different methods.

Regarding focal activity, the scalp distributions on which vMMN focal activity may be less visible are the 200-270 ms in older group for happy faces and the 150-205 ms in the older group for angry faces. In both cases there is a frontal negativity that may obscure a posterior negativity; however, note that the anterior and posterior negativity seem to be unrelated in the case of the 200-270 ms interval as there is a “drop” in the central regions and that there is focal activity at PO3 and PO7 in the 150-205 ms interval which suggests posterior negativity independent of the anterior one. All remaining scalp distributions show focal activity at expected electrode sites for vMMN to face stimuli. We have edited the figures to make the scalp distributions more visible.

We edited Figures 2-4 by colouring the ranges of significant t-tests in addition to the frames and enhanced those intervals. The time of the ranges (e.g., 90-161 ms) is available under the scalp distributions.

5. Finally, the Discussion is meager. There is very little discussion of (what I see) as a lack of difference in the vMMNs between age groups. Moreover, the discussion of the age-group differences that did occur is extremely limited and, of necessity, completely post-hoc. For example, I do not see why an N400 would be expected in the oddball situation presented in the paper. Yet the authors do very little by way of telling us why an N400 might have occurred and what type of cognition that might entail for the older relative to the younger adults who produced an early positivity. The authors do little in the way of interpreting these age-related differences. Why would they occur and what do they mean in terms of the differences in cognition (if they exist here) between the young and older adults?

- We are grateful to the reviewer for drawing attention to the unclear points and the insufficient detail in this part of the Discussion! We used N400 as an example of what processes may be involved at this stage of processing; however, we were not clear about regarding the negativity to be akin to N400 but not N400 itself. We have now expanded and modified the text as follows:

“One study [44] found a difference to deviant faces of old models among standard faces of young models in the 330-370 ms range in older adults, which the authors interpreted as a delayed vMMN. However, this interpretation remains debated, as responses in this latency range may reflect higher-level evaluative processes rather than classical mismatch detection. In the present study we did not observe a comparable delayed response to emotional faces in the emotional-face-only condition. One possibility is that violations involving more complex or abstract associations (e.g., age in [44], or cross-category associations in the present study) may not elicit a canonical vMMN in older adults, but instead engage later processing stages. At this stage of stimulus processing, it is likely that mechanisms other than predictive coding are involved. Indeed, the detection of incongruence in this latency range is well documented in other domains, most prominently in the N400 component, which is typically observed in response to unexpected verbal stimuli (for a review see [45]).”

We added the following text that refers to the difference between the age groups and is also a includes modifications based on a comment by Reviewer #3:

“The early positivity has been interpreted as reflecting feature-specific, memory-based change detection, possibly related to the vMMN [41–43], i.e , in the younger group an early mismatch between expected features was sufficient to signal a violation of the association. Another important consideration is the nature of the manipulated regularity. VMMN in the context of complex stimuli remains largely unexplored (e.g., [47]). Unlike classical vMMN paradigms, the scene-emotional face pair condition involved category-level alternations between complex stimuli, which may not give rise to stable low-level predictions required for eliciting a classical vMMN. These alternations may engage higher-level categorical or semantic processing, not captured by early vMMN components. This may be reflected in the later negativity in older adults, where the association appears to have been encoded as a more complex regularity.”

6. On another note, it might have been useful to determine if there was any association, across participants, between the behavioral data and the size of the vMMN, the positivity in young adults and the "N400" in older adults. These associations, were they present and differed between young and older adults, might have aided the authors in coming up with a more definitive explanation for the age-related differences that did occur.

- We agree with the reviewer that in EEG studies with behavioural data, it is sensible to look for correlations between behaviour and brain signal. However, we would like to emphasize that correlating behavioural performance with the vMMN is not theoretically well motivated in the present paradigm. By definition, the vMMN reflects automatic change detection elicited by task-irrelevant stimuli, and it is specifically designed to be independent of task performance.

In our experiment, behavioural responses were required only for the task-relevant stimulus (frame colour change), while the vMMN was elicited by entirely independent, task-irrelevant stimuli (scenes and faces). As a consequence, any association between behavioural measures and vMMN amplitude would not reflect a functional brain–behaviour relationship, but rather an incidental correlation that is difficult to interpret theoretically. Neither of the observed components in either condition in the study was elicited by the task-relevant stimulus stream, and therefore they cannot be meaningfully linked to task performance.

Note that any epoch containing a task-irrelevant stimulus that was within 800 ms of a frame colour change and/or within 800 ms of a response was removed from the ERP analysis. The 800-ms time interval is a sensible option for a simple reaction time task.

Note: While performing an additional analysis for RT by BLOCK, we discovered that our script had mislabelled some of the trials – approximatel

---

## [Decision Letter · Decision Letter 1]

24 Feb 2026

Automatic pairing of real-world stimuli in younger and older adults: An event-related potential study

PONE-D-25-41490R1

Dear Dr. Kojouharova,

We’re pleased to inform you that your manuscript has been judged scientifically suitable for publication and will be formally accepted for publication once it meets all outstanding technical requirements.

Kind regards,

Árpád Csathó, Ph.D.

Academic Editor

PLOS One

Additional Editor Comments (optional):

Reviewers' comments:

Reviewer's Responses to Questions

**Comments to the Author**

Reviewer #2: All comments have been addressed

Reviewer #3: All comments have been addressed

2. Is the manuscript technically sound, and do the data support the conclusions?

Reviewer #2: Yes

Reviewer #3: Yes

3. Has the statistical analysis been performed appropriately and rigorously?

Reviewer #2: Yes

Reviewer #3: Yes

4. Have the authors made all data underlying the findings in their manuscript fully available?

Reviewer #2: Yes

Reviewer #3: Yes

5. Is the manuscript presented in an intelligible fashion and written in standard English?

Reviewer #2: Yes

Reviewer #3: Yes

Reviewer #2: (No Response)

Reviewer #3: I believe the authors have satisfactorily addressed all the issues raised, particularly the additional analysis of the ERP data and the interpretation of results in the Discussion.

**Do you want your identity to be public for this peer review?** For information about this choice, including consent withdrawal, please see our Privacy Policy

Reviewer #2: No

Reviewer #3: No

---

## [Editor Report · Acceptance letter]

PONE-D-25-41490R1

PLOS One

Dear Dr. Kojouharova,

I'm pleased to inform you that your manuscript has been deemed suitable for publication in PLOS One. Congratulations! Your manuscript is now being handed over to our production team.

Kind regards,

on behalf of

Dr. Árpád Csathó

Academic Editor

PLOS One